# Improved I_on_/I_off_ Current Ratio and Dynamic Resistance of a p-GaN High-Electron-Mobility Transistor Using an Al_0.5_GaN Etch-Stop Layer

**DOI:** 10.3390/ma15103503

**Published:** 2022-05-13

**Authors:** Hsiang-Chun Wang, Chia-Hao Liu, Chong-Rong Huang, Min-Hung Shih, Hsien-Chin Chiu, Hsuan-Ling Kao, Xinke Liu

**Affiliations:** 1College of Materials Science and Engineering, Shenzhen University-Hanshan Normal University Postdoctoral Workstation, Shenzhen University, Shenzhen 518060, China; smallflgt@hotmail.com; 2Key Laboratory of Optoelectronic Devices and Systems, Ministry of Education and Guangdong Province, College of Physics and Optoelectronic Engineering, Shenzhen University, Shenzhen 518060, China; xkliu@szu.edu.cn; 3Department of Electronic Engineering, Chang Gung University, Taoyuan 333, Taiwan; r3287133@gmail.com (C.-H.L.); gain525252@gmail.com (C.-R.H.); h27024007@gmail.com (M.-H.S.); snoopy@mail.cgu.edu.tw (H.-L.K.); 4Department of Radiation Oncology, Chang Gung Memorial Hospital, Taoyuan 333, Taiwan

**Keywords:** etch-stop layer, high selectivity ratio, normally off, p-GaN gate HEMT

## Abstract

In this study, we investigated enhance mode (E-mode) p-GaN/AlGaN/GaN high-electron-mobility transistors (HEMTs) with an Al_0.5_GaN etch-stop layer. Compared with an AlN etch-stop layer, the Al_0.5_GaN etch-stop layer not only reduced lattice defects but engendered improved DC performance in the device; this can be attributed to the lattice match between the layer and substrate. The results revealed that the Al_0.5_GaN etch-stop layer could reduce dislocation by 37.5% and improve device characteristics. Compared with the device with the AlN etch-stop layer, the p-GaN HEMT with the Al_0.5_GaN etch-stop layer achieved a higher drain current on/off ratio (2.47 × 10^7^), a lower gate leakage current (1.55 × 10^−5^ A/mm), and a lower on-state resistance (21.65 Ω·mm); moreover, its dynamic R_ON_ value was reduced to 1.69 (from 2.26).

## 1. Introduction

In recent years, the wide bandgap (WBG) GaN-based high-electron-mobility transistors (HEMTs) have attracted much attention for high-radio-frequency (RF) and high-power semiconductor device applications owing to their excellent performance of high electric field strength (3.3 MV/cm), high mobility (>1200 cm^2^/Vs), and favorable thermal conductivity [1,2,3]. To get realistic power supply applications, the normally off behavior of GaN-based HEMTs must be implemented. Thus, several methods have attempted to realize the positive threshold voltage (V_TH_) of AlGaN/GaN HEMTs, such as ultrathin barriers, gate-recessed structures, fluorine treatment, and p-type gates [4,5,6,7,8]. However, structures involving p-type gates have drawn increasing attention in industry owing to their low on-state resistance and high threshold voltage. The precision and lower damage etching process is a key factor in device fabrication, and outstanding etching depth control is imperative because the residual p-GaN layer in the out-of-gate area makes the 2DEG channel depleted and leads to a low forward current. Moreover, if the p-GaN layer is overetched in to the AlGaN barrier layer and reduces the AlGaN barrier thickness in this process, the channel carrier density is decreased because of the decrease in spontaneous polarization. In tradition structures, an AlN layer is used as an etch-stop layer; however, high-quality thin AlN has difficulty achieving epitaxy control, and problems related to lattice mismatch can arise [9,10]. In this study, we applied an Al_0.5_GaN etch-stop layer between p-GaN and AlGaN barriers and compared its performance with that of an AlN etch-stop layer. The experimental results indicated that the dynamic R_ON_ was improved and that the leakage current was suppressed. Transmission electron microscopy (TEM) images revealed the etch-stop layer to be smooth and highly selective. Therefore, high-performance normally off p-GaN/AlGaN/GaN HEMTs can be realized using an AlGaN etch-stop layer for high-speed and high-power electronic applications.

## 2. Device Structure

As illustrated in Figure 1a, we present a new p-GaN/AlGaN/GaN HEMT structure with an Al_0.5_GaN etching stop design grown through metal organic chemical vapor deposition (MOCVD) on a 6 in silicon (111) wafer. The epistructure was composed, from bottom to top, of a 4 μm-thick C-doped GaN buffer layer, a 300 nm-thick undoped GaN channel layer, a 12 nm-thick undoped Al_0.17_GaN barrier layer, a 2 nm Al_0.5_GaN etch-stop layer, and a 75 nm-thick p-type GaN top layer whose active Mg concentration was 1 × 10^18^ cm^−3^.

For device fabrication, we used standard photolithography and lift-off technology, the active region was defined by a photoresist and etched to a depth of 200 nm using BCl_3_/Cl_2_ mixed-gas plasma by reactive ion etching (RIE). A 5 μm-long p-type GaN gate platform was formed through a mixture of BCl_3_/Cl_2_/SF_6_ gas plasma 120 s to remove p-GaN by inductively coupled plasma (ICP). The F radicals from the SF_6_ plasma and Al atoms from the Al_0.5_GaN barrier layer created a fluorination reaction and formed a thin aluminum fluoride (AlF_3_) etch-stop layer [11]. The resulting etching rate was 0.625 nm/s. Moreover, the A_0.5_GaN layer was etched at an etching rate of <0.016 nm/s, implying a high-selectivity etching process in the p-GaN/Al_0.5_GaN layer. Subsequently, the formed AlF_3_ can be removed by diluted HF/NH_4_OH chemical solution [12]. As indicated in Figure 1b the p-GaN removal depth was measured using an atomic force microscope (AFM) and the inset of that figure presents a TEM image after p-GaN etching. This was followed by the source and drain ohmic contact formation where a Ti/Al/Ni/Au (25 nm/120 nm/25 nm/150 nm) ohmic metal stack was deposited by electron beam evaporation and thermally annealed at 875 °C for 30 s in ambient nitrogen (N_2_) by rapid thermal annealing system (RTA). Third, the device was fabricated with implant isolation through oxygen implantation. Finally, the Ni/Au (25 nm/120 nm) gate electrode (gate length: 2 μm) was deposited through electron beam evaporation, and 100 nm of SiN was passivated.

## 3. Experimental Results and Discussion

X-ray diffraction (XRD) was used to investigate the dislocation density, and the results are presented in Figure 2. The full width at half-maximum (FWHM) values of the (002) symmetric and (102) asymmetric reflection were used to measure crystal quality. In our device surface measurements, we mainly investigated the crystal quality on the device surface. The FWHM values for the (002) and (102) planes of the AlN and AlGaN etch-stop layer designs were 164/239 and 162/179 arcsec, respectively. In general, the rocking curve scan of a (002) reflection provides information on the degree of tilt with respect to the surface of a device, and the FWHM of this reflection is a qualitative measure of screw dislocation density (*N_screw_*) [13,14]. The rocking curve scan of a (102) reflection provides information on the degree of twist with respect to the surface of a device, and the FWHM of this reflection is a measure of edge dislocation density (*N_edge_*). The dislocation density can be calculated using XRD-derived FWHM results as follows:(1)$$N_{screw}=\frac{FWHM_{002}^{2}}{4.35\times b_{screw}^{2}},N_{edge}=\frac{FWHM_{102}^{2}}{4.35\times b_{edge}^{2}},\;\;$$
(2)$$N_{total}=N_{screw}+N_{edge}$$
where *N_screw_* and *N_edge_* are the screw and edge dislocation densities, respectively, and b is Burger’s vector. In this study, these equations were used for calculation, and the results revealed that the total dislocation (*N_total_*) values of the Al_0.5_GaN etch-stop layer and AlN etch-stop layer were 2.23 × 10^8^/cm^2^ and 3.57 × 10^8^/cm^2^, respectively. As indicated in Figure 2, the screw dislocation density and edge dislocation density were lower when the Al_0.5_GaN etch-stop layer was used than when the AlN etch-stop layer was used.

To study the effects of the layers on DC performance, we measured the transfer (*I_DS_*–*V_GS_*) and output (*I_DS_*–*V_DS_*) characteristics of the devices by using an Agilent 4142B monitor. Figure 3a presents a plot of the log-scale *I_DS_*–*V_GS_* curve as a function of gate-to-source voltage (*V_GS_*) under biasing at a drain-to-source voltage (*V_DS_*) of 10 V. The threshold voltage (*V_TH_*) of the devices was defined as the *V_GS_* level at which I_DS_ reached 1 mA/mm. We observed that threshold voltage (*V_TH_*) values of the devices with the AlN etch-stop layer and AlGaN etch-stop layer were 0.37 and 0.23 V, respectively. At a gate bias of 4 V, the maximum output current density (*I_Dmax_*) values of the devices with the AlN and AlGaN etch-stop layers were 67 and 121 mA/mm, respectively. At a gate bias of −1 V, the off-state current values of the devices with the AlN and AlGaN etch-stop layers were 4.31 × 10^−5^ and 4.11 × 10^−6^ mA/mm, respectively. The device with the Al_0.5_GaN etch-stop layer had a superior *I*_on_/*I*_off_ ratio to the device with the AlN etch-stop layer, which increased from 1.41 × 10^6^ to 2.47 × 10^7^. Moreover, the subthreshold swing (S.S.) values of the devices with the AlN and Al_0.5_GaN etch-stop layers were 103.5 and 99.2 mV/dec, respectively. The subthreshold swing (S.S.) is expressed by the analytical equation given below [15]
(3)$$S.S.=\left(\ln10\right)\left(\frac{kT}{q}\right)\left(1+\frac{Cd+Cit}{Cox}\right)\;$$
(4)$$Dit=\frac{Cit}{q}\;$$
where *kT*/*q* is the thermal voltage, *C_ox_* is the capacitance of the gate dielectric, *C_d_* the depletion capacitance, *C_it_* is the capacitance of gate and semiconductor interface state, *D_it_* is interface charge densities, and *q* is the electronic charge. The capacitance of the gate dielectric (*C_ox_*) for the AlN stop layer and Al_0.5_GaN stop layer was measured to be 222 nF/cm^2^ and 184 nF/cm^2^ under a frequency of 1M Hz. Therefore, the interface charge densities (Dit) can be calculated and the *D_it_* values of the AlN stop layer and Al_0.5_GaN stop layer device were 1.02 × 10^12^ and 7.64 × 10^11^ cm^−2^ eV^−1^, respectively. This indicated the device with the Al_0.5_GaN stop layer had better defect density suppression. The Al_0.5_GaN stop layer design was determined to be suitable for device switching owing to its favorable *I*_on_/*I*_off_ ratio and gate drive-control capability. Figure 3b illustrates the *I_DS_*–*V_DS_* curves as functions of the gate-to-source voltage (*V_GS_*) bias ranging from 0 to 4 V in steps of 1 V and of the drain-to-source voltage (*V_DS_*) sweep ranging from 0 to 10 V. The ON-resistance (*R_ON_*) could be reduced from 28.03 to 21.65 Ω mm owing to the lower dislocation trap density and the suppressed trap density effect in the channel.

The gate leakage curve presented in Figure 4a was used to investigate the leakage current mechanism. The device with the Al_0.5_GaN etch-stop layer exhibited a lower gate leakage current than did the other device. This low gate leakage current not only increased the device breakdown voltage but also improved the gate operator voltage. The off-state breakdown voltage (*V_BR_*) was measured using an Agilent B1505 analyzer; the *V_GS_* was 1V and the drain leakage current reached 1 mA/mm. As displayed in Figure 4b, the *V_BR_* values of the devices with the AlN and Al_0.5_GaN etch-stop layers were 501 and 561 V, respectively. Moreover, Baliga’s figure of merit (*BFOM* = *V_BR_*^2^/*R_DS_*__*on*_) for various power transistors was calculated to evaluate the overall performance of these devices [16,17]. The *BFOM* values of the devices with the AlN and Al_0.5_GaN etch-stop layers were 44.37 and 83.11 MW/cm^2^, respectively.

The Maury AMCAD pulse IV system was used to further investigate trapping/detrapping phenomena and the dynamic behavior of the devices [18,19]. Furthermore, the *I_DS_*–*V_DS_* characteristics were also measured from different quiescent bias points at *V_GS_* = 4 V to investigate the influence of off-state gate bias stress on dynamic *R_ON_* and *I_DS_*, as illustrated in Figure 5. The reference off-state was set to (*V_GSQ_*, *V_DSQ_*) = (0 V, 0 V); this setting did not induce any relevant trapping. The device was switched with a 2 μs pulse width and 200 μs period. The quiescent gate bias (*V_GSQ_*) was swept from 0 to −3 V. The current collapse in the device with the AlN etch-stop layer was worse than that in the device with the Al_0.5_GaN etch-stop layer, and the high gate lag of the device with the AlN layer under gate voltage stress resulted in a decrease in the *I*–*V* slope, indicating that the surface defect trap density of this device was higher than that of the device with the Al_0.5_GaN etch-stop layer. The dynamic *R_ON_* (*R_ON_*/*R_ON_*_(0,0)_) of the device with the Al_0.5_GaN etch-stop layer slightly increased with the gate bias stress from 0 to −3 V because of the low electron injection into the surface trap states from the gate electrode [20]. The dynamic *R_ON_* ratio increased to 2.26 and the dynamic drain current decreased to 41.4% when the off-state gate bias stress was −3 V for the device with the AlN etch-stop layer.

## 4. Conclusions

In this study, a highly selective Al_0.5_GaN etch-stop layer was applied to a p-GaN/AlGaN/GaN HEMT. Compared with the traditional AlN etch-stop layer structure, the Al_0.5_GaN etch-stop layer had a lower dislocation density, according to our XRD results; dynamic *R_ON_* and dynamic I_DS_ were significantly lower in the device with the Al_0.5_GaN etch-stop layer. Furthermore, the device with the Al_0.5_GaN etch-stop layer had superior DC characteristics to the other device, including a lower off-state current, lower gate leakage, lower on-resistance, higher on/off ratio, good subthreshold swing, and higher off-state breakdown voltage. A *BFOM* assessment revealed that applying an Al_0.5_GaN etch-stop layer is a promising method for fabricating high-performance normally off p-GaN HEMTs.

## Figures and Tables

**Figure 1 materials-15-03503-f001:**
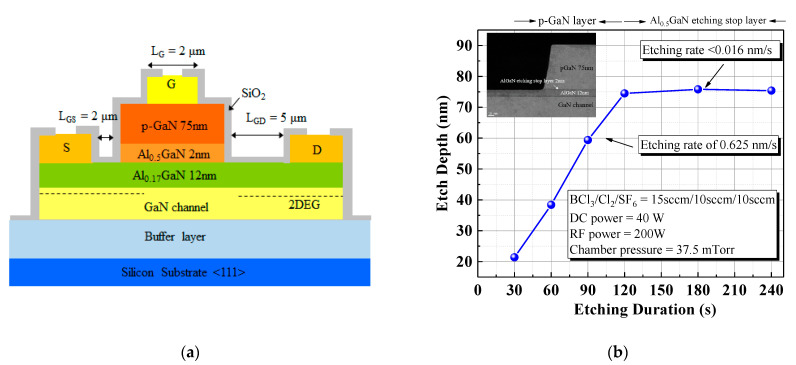
(**a**) Schematic cross-sectional structure of the p-GaN gate HEMT with an Al_0.5_GaN etch-stop layer design. (**b**) p-GaN etch depth versus the etch duration and TEM image after etching (inset).

**Figure 2 materials-15-03503-f002:**
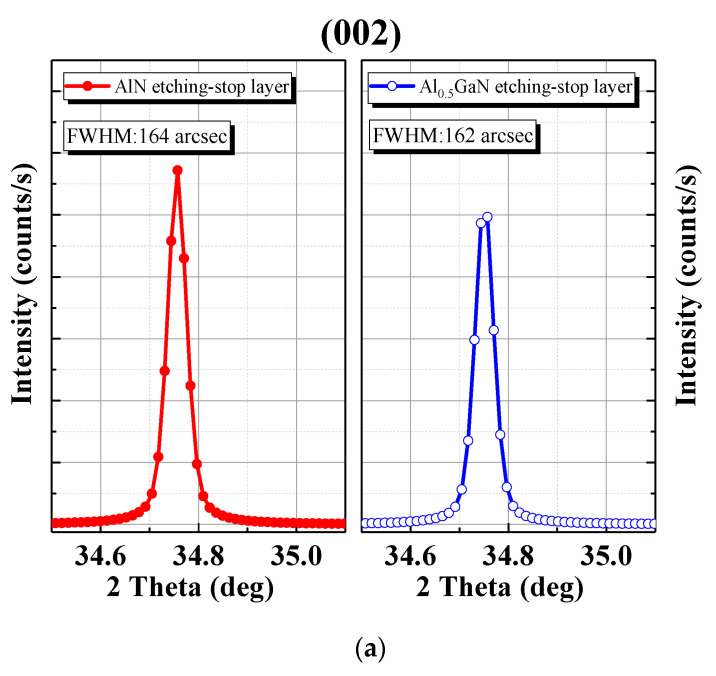
XRD measurement of the FWHM at the Al_0.5_Ga N etch-stop layer and AlN etch-stop layer. (**a**) (002) symmetric reflection and (**b**) (102) asymmetric reflection.

**Figure 3 materials-15-03503-f003:**
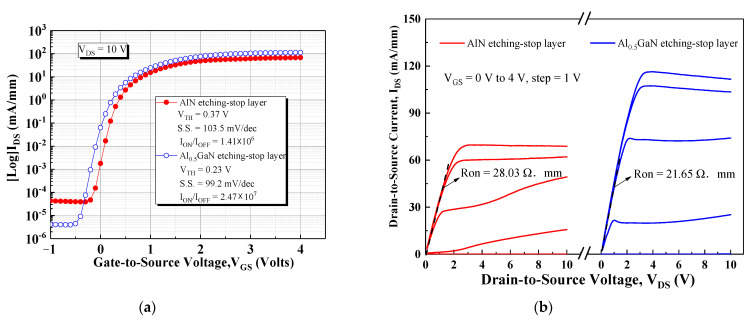
I–V characteristic comparison of both devices (device dimension: L_GS_/L_G_/L_GD_/W_G_ = 2 μm/2 μm/5 μm/100 μm). (**a**) Transfer characteristics and (**b**) output characteristics.

**Figure 4 materials-15-03503-f004:**
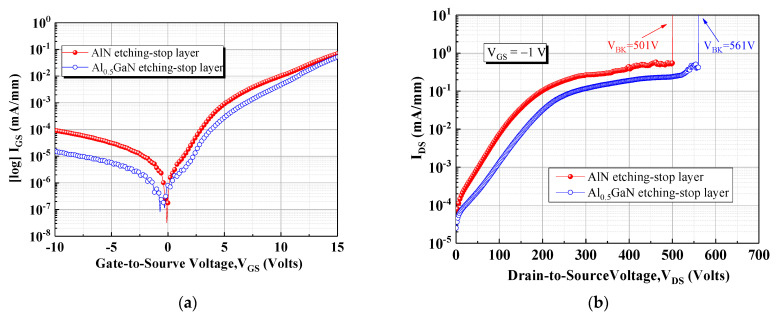
(**a**) Gate leakage characteristics and (**b**) off-state breakdown voltage measurement of both devices.

**Figure 5 materials-15-03503-f005:**
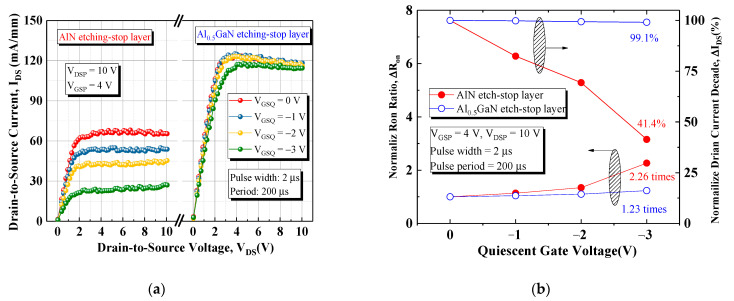
(**a**) Pulsed *I_DS_*–*V_DS_* characteristics after quiescent gate bias (*V_GSQ_*) stress and (**b**) dependence of the *R_ON_* collapse ratio and I_DS_ decay versus various quiescent gate voltages of both devices.

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
