# Peer review of "Improved I_on_/I_off_ Current Ratio and Dynamic Resistance of a p-GaN High-Electron-Mobility Transistor Using an Al_0.5_GaN Etch-Stop Layer"

_materials, 2022, doi:10.3390/ma15103503_

Round 1
Reviewer 1 Report
The content and structure of the thesis seems appropriate. However, it seems that a few minor modifications are needed below.
- When writing an abbreviation, the full name must be written first. In the introduction, VTH should be corrected to the threshold voltage (VTH).
- Proper reference for explanation of AlN is required. It is recommended to add one or more references at the end of the sentence “high quality thin AlN has difficulty achieving epitaxy control, and problems related to lattice mismatch can arise” in the introduction.
- In line 110, change dc to uppercase.
Author Response
- When writing an abbreviation, the full name must be written first. In the introduction, VTH should be corrected to the threshold voltage (VTH).
Ans: Thanks for your careful reading and we have revised this typo.
- Proper reference for explanation of AlN is required. It is recommended to add one or more references at the end of the sentence “high quality thin AlN has difficulty achieving epitaxy control, and problems related to lattice mismatch can arise” in the introduction.
Ans: Thanks for your kindly suggestion. We added two reference [9-10] for explanation of AlN layer.
[9] W Tian et al. Effect of growth temperature of an AlN intermediate layer on the growth mode of AlN grown by MOCVD. J. Phys. D: Appl. Phys. 2013, 46, 065303.
[10] I Demir et al Sandwich method to grow high quality AlN by MOCVD. J. Phys. D: Appl. Phys.2018, 51, 085104.
3.In line 110, change dc to uppercase.
Ans: Thanks for your careful reading and we have revised this typo.

Reviewer 2 Report
In the manuscript, the benefits coming from the replacement of etch-stop layer in high-electron-mobility transistors were presented. Are there any drawbacks coming from the above modification of the structure? Does it have any impact on the etching process? How about the thickness of this layer? Did you investigated the impact of the thickness of etch-stop layer on the device characteristics? Can we consider 2 nm as an optimal thickness? I suggest to place the answers in manuscript to at least some of the above questions.
In "3. Experimental Results and Discussion", the reader may find the discussion related to the obtained results. However, it is mainly about the data that may be extracted from the figures, and the discussion about the physical phenomena is relatively short. Can we therefore formulate the conclusion that the better results come from the reduced number of lattice defects which dominates the effects related to using the ternary material with different thermal and electrical properties?
Author Response
- In the manuscript, the benefits coming from the replacement of etch-stop layer in high-electron-mobility transistors were presented. Are there any drawbacks coming from the above modification of the structure? Does it have any impact on the etching process? How about the thickness of this layer? Did you investigated the impact of the thickness of etch-stop layer on the device characteristics? Can we consider 2 nm as an optimal thickness? I suggest to place the answers in manuscript to at least some of the above questions.
Ans: Thanks for your kindly suggestion. Due to etching stop layer usually used the high Al content and the increased Al content in AlGaN leaded to the lattice mismatch and Dx trapping center. The thickness was controlled very thin and ensured the etching stop function, therefore the Al0.5GaN of 1 -2nm thickness was suitable and general. In addition, reduced Al content in tradition AlN etching stop layer and using Al0.5GaN etching stop layer can improve lattice mismatch. Please noted that the process is easy to over etching when the decreased Al content, so the suitable etch selection ratio was very important by etching parameter adjustment. In this study, we controlled the reactive gas and power of process to achieve high etch selection ratio.
In "3. Experimental Results and Discussion", the reader may find the discussion related to the obtained results. However, it is mainly about the data that may be extracted from the figures, and the discussion about the physical phenomena is relatively short. Can we therefore formulate the conclusion that the better results come from the reduced number of lattice defects which dominates the effects related to using the ternary material with different thermal and electrical properties?
Ans: Thanks for your kindly suggestion. Due to the GaN is heterostructure, so lattice mismatch and dislocation are existing issues. Indeed, the lattice defects effect mainly cause different thermal and electrical properties at same process. In this paper, Al0.5GaN of 1 -2nm thickness was controlled in critical thickness with extremely low strain effect. Therefore, thermal and electrical proprieties variation can be minor issues. But we will still further check and study these parameters in the near future.

Reviewer 3 Report
The authors submitted a manuscript dealing with the high electron mobility transistor (HEMT) fabrication technology development. Presented data demonstrate plausible results, and my only comment aims to improve the manuscript quality. A more detailed defect density analysis should be proper since the technology optimisation is done for defect density suppression.
I strongly recommend to include interface state density evaluation using the subthreshold swing (SS) as SS=kT/e*ln(10)*(1-eN/C), where kt/e is the thermal voltage, C is the gate capacitance per unit of area, and N is the interface charge density. Calculated value should be compared with defect density estimated by the x-ray diffraction (XRD).
As a result, I strongly support the manuscript, and my only recommendation aims on the manuscript improvement.
Author Response
The authors submitted a manuscript dealing with the high electron mobility transistor (HEMT) fabrication technology development. Presented data demonstrate plausible results, and my only comment aims to improve the manuscript quality. A more detailed defect density analysis should be proper since the technology optimization is done for defect density suppression.
I strongly recommend to include interface state density evaluation using the subthreshold swing (SS) as SS=kT/e*ln(10)*(1-eN/C), where kt/e is the thermal voltage, C is the gate capacitance per unit of area, and N is the interface charge density. Calculated value should be compared with defect density estimated by the x-ray diffraction (XRD).
As a result, I strongly support the manuscript, and my only recommendation aims on the manuscript improvement
Ans: Thanks for reviewer’s suggestion. From subthreshold swing (SS) is expressed by the analytical equation given below
where kT/q is the thermal voltage, Cox is the capacitance of the gate dielectric, Cd the depletion capacitance, Cit is the capacitance of gate and semiconductor interface state, Dit is Interface charge densities, q is the electronic charge. The capacitance of the gate dielectric (Cox) for AlN stop layer and Al0.5GaN stop layer is measured to be 222 nF/cm2 and 184 nF/cm2 under a frequency of 1M Hz. Therefore, the interface charge densities (Dit) can be calculated that the Dit of AlN stop layer and Al0.5GaN stop layer device were 1.021012 and 7.64 1011 cm-2eV-1, respectively. This indicates the device of Al0.5GaN stop layer had better defect density suppression.
Above results, we have added in in the revised version.
